# Improving Human Diets and Welfare through Using Herbivore-Based Foods: 1. Human and Animal Perspectives

**DOI:** 10.3390/ani14071077

**Published:** 2024-04-02

**Authors:** John R. Caradus, David F. Chapman, Jacqueline S. Rowarth

**Affiliations:** 1Grasslanz Technology Ltd., PB 11008, Palmerston North 4442, New Zealand; 2DairyNZ, Lincoln 7647, New Zealand; david.chapman@dairynz.co.nz; 3Faculty of Agriculture and Life Science, Lincoln University, 85084 Ellesmere Junction Road, Lincoln 7647, New Zealand; jsrowarth@gmail.com

**Keywords:** animal-based food, animal health, nutrition, plant-based food

## Abstract

**Simple Summary:**

Optimal human health requires the adequate provision of all nutrients in the correct proportions, ensuring the provision of energy and essential small molecules. All primates, including humans, are omnivorous, but our most striking difference from other primates is the remarkable diversity of diet we consume. The aim here is to examine the benefits and possible unintended consequences of using herbivore-based foods on human and animal health and welfare. The advantages of using grazed pasture for ruminant meat and milk production include (1) animal-sourced foods contain essential amino acids and micronutrients, and almost always have a higher digestibility than plant proteins, (2) greater use of pasture to feed animals could result in less use of food that could be used for human consumption, e.g., maize, soybean, and cereal crops, (3) ruminants can provide food from land otherwise unable to be used for cropping, consume feed that cannot be eaten directly by humans, and provide more than just food—they also provide leather and fibre. Animal-sourced foods are an important part of the human diet, and while some unintended consequences associated with animal health and herd management have occurred, technologies and systems to provide solutions to these are available and under refinement.

**Abstract:**

Human health and diet are closely linked. The diversity of diets consumed by humans is remarkable, and most often incorporates both animal and plant-based foods. However, there has been a recent call for a reduced intake of animal-based foods due to concerns associated with human health in developed countries and perceived impacts on the environment. Yet, evidence for the superior nutritional quality of animal-sourced food such as meat, milk, and eggs, compared with plant-based foods, indicates that consumption of animal-sourced food should and will continue. This being the case, the aim here is to examine issues associated with animal-sourced foods in terms of both the quantification and mitigation of unintended consequences associated with environment, animal health, and herd management. Therefore, we examined the role of animal proteins in human societies with reference to the UN-FAO issues associated with animal-sourced foods. The emphasis is on dominant grazed pastoral-based systems, as used in New Zealand and Ireland, both with temperate moist climates and a similar reliance on global markets for generating net wealth from pastoral agricultural products. In conclusion, animal-sourced foods are shown to be an important part of the human diet. Production systems can result in unintended consequences associated with environment, animal health, and herd management, and there are technologies and systems to provide solutions to these that are available or under refinement.

## 1. Introduction

A prerequisite for optimal human health is the adequate provision of all nutrients in the correct proportions, ensuring the provision of energy and essential small molecules that the body cannot synthesise, and the creation of structure [1]. Whereas humans are omnivorous, as are all primates, our most striking difference from other primates is the remarkable diversity of diet we consume. This is referred to as dietary plasticity which has developed through “cultural and technological innovations for processing resources, and which has allowed human populations to expand into new and more marginal ecosystems” [2,3]. This has also resulted in genetic variation within a population, upon which natural selection can operate, resulting in differences in nutrient tolerances and intolerances [4] and perhaps preferences. A review of 15 studies demonstrated that “the relative contribution of animal foods varies considerably, ranging from less than 20% of dietary energy in traditional farming communities of tropical South America, to more than 95% among traditionally living Inuit hunters of the Canadian Arctic” [3]. Globally, livestock products have been estimated to supply about 17% of calorie consumption and 33% of protein consumption for humans [5]; or alternatively, 11% of food energy availability and 21% of protein [6]. Livestock prsoduction occupies about a third of all ice-free land globally [7], and has been estimated to contribute about 8 to 11% [8], or even as high as 30% [9,10] of global anthropogenic greenhouse gas emissions. A recent call has been made for “strategic investment in research and development around plant-based foods, greening public procurement practices and ensuring national dietary guidelines taking sustainability into account” [11]. Whereas plant-based sources currently provide 57% of protein consumed by humans globally [12,13], the extent of the development and uptake of animal-sourced food alternatives (novel plant-based food, cultivated meat, and fermentation-derived products) is under debate [14,15,16,17] as to whether these will complement rather than replace conventional animal-sourced foods [18]. Significant replacement is unlikely, considering that estimates of population growth by 2050 would result in demand for food, feed, and fibre around the globe to increase by 51% [19].

Worldwide, over 80% of food production is derived from just 11 crop species, of which about two-thirds are cereals [20]. Even in higher income, ‘western’ countries, where only about a third of protein intake comes directly from plant sources, considerable amounts of grain are used to feed animals to produce essential protein and fats for human consumption. Increased use of plant proteins in human diets has been proposed as a means to satisfy dietary requirements for an increasing world population [21] and improve nutrient use efficiency [22]. Indeed, it has been estimated that “the current production of crops is sufficient to provide enough food for the projected global population of 9.7 billion in 2050, although very significant changes to the socio-economic conditions of many (ensuring access to the global food supply) and radical changes to the dietary choices of most (replacing most meat and dairy with plant-based alternatives, and greater acceptance of human-edible crops currently fed to animals, especially maize, as directly-consumed human food) would be required” [23]. It has been concluded [24] “that diets and livestock feed intake are of prime importance for future GHG emissions from the food system, while raising crop yields does not necessarily result in low emissions”. However, it has been calculated that although moving American agriculture to plant-only would increase food supply by 23% and decrease greenhouse gas emissions by 28%, it would meet fewer of the US population’s requirements for essential nutrients [25]. An evaluation of nutritional adequacy using least-cost diets produced from foods available indicated that the result of a plant-only diet would have more nutrient deficiencies, a greater excess of energy, and a need to consume a greater amount of food solids (i.e., potential weight gain). The assessment suggested the reduction in greenhouse gas emissions achieved by removing animals from US agriculture would be associated with a food supply incapable of supporting the US population’s nutritional requirements. Despite this assessment, it has been proposed that “eliminating the use of animals as food technology would produce substantial negative emissions [substantial emission reduction] of all three major greenhouse gases” (carbon dioxide, methane, and nitrous oxide) [26]. The authors proposed that even though animal-based products contribute significant amounts of energy and protein “they are not necessary to feed the global population”, and that “existing crops could replace the calories, protein and fat from animals with a vastly reduced land, water, GHG and biodiversity impact, requiring only minor adjustments to optimize nutrition”. In support, they reference Springmann et al. [27], but that study concluded that “approaches for sustainable diets [from animal sourced to entirely plant based] are context specific and can result in concurrent reductions in environmental and health impacts globally and in most regions, particularly in high-income and middle-income countries, but they can also increase resource use in low-income countries when diets diversify”.

Another factor to consider is the extent to which food for human consumption can be derived from complementary rather than competitive plant sources by feeding plant byproduct-based concentrates instead of human-edible, plant-based feed to ruminants [28]. It has been demonstrated that less human-edible plant feed is needed in ruminant systems than in monogastric systems (6 vs. 16 kg of human-edible feed dry matter per kilogram of protein in meat, milk, or eggs produced) [29]. It has been estimated that if animals are fed only from grassland and byproducts from food production, then by 2050, there could be reductions in greenhouse gas emissions of −18%; arable land occupation −26%, N-surplus −46%; P-surplus −40%; non-renewable energy use −36%, pesticide use intensity −22%, freshwater use −21%, and soil erosion potential −12% [30]. Currently, 13% of global beef production is from grain fed animals; thus, 87% comes from feeds that cannot be consumed directly by humans [29].

A further dimension to the dilemma of balancing global food supply with environment and other imperatives are the global trends in agricultural output and productivity (the latter taking into account the use of the key inputs required to produce food), and the effects of global climate change on those trends. Such analyses are notoriously difficult to perform due to, among other factors, problems in estimating the quantity of inputs such as labour and capital [31]. Nonetheless, Fuglie [32] found no evidence of the global rate of growth of agricultural land, labour, or total factor productivity having slowed from 1960 to 2014. However, there was “evidence that agricultural output and productivity growth has slowed in industrialised countries”, but has been offset by accelerated rates of gain in developing countries. This was specifically in cropping, where increased intensity of use of crop land (more harvests per year) outweighed reductions in crop yield per harvest. Fuglie [32] also noted that, globally, “growth rates in overall animal productivity have slowed”, despite gains in non-ruminant species (pigs and poultry) from improved feed conversion efficiency. More recently, and in contrast to Fuglie [32] in some regards, Ortiz-Bobea et al. [33] reported that anthropogenic climate change is already substantially eroding rates of agricultural productivity growth. This trend was most severe in regions where food security is already uncertain. For example, rates of total factor agricultural productivity growth in sub-Saharan Africa, North Africa, and near East and Latin America were estimated to have declined by between 26 and 34% since approximately 1960 compared with the rates of gain expected from a climate system unaffected by human-induced climate change [33]. Most of the signals emerging from these analyses point to static or declining overall global productivity growth, with some authors noting associations with an increase in the rate at which labour is leaving the agricultural sector [32], and markedly lagging rates of public investment in agricultural research and development in high-income countries in recent decades [34].

In animal production, as humankind moved from being hunter–gatherers to a more sedentary and organised farming paradigm, people became reliant on domesticated forages, crops, and animals [35]. More recently, as land for farming has become less available, the solution has been to bring conserved fodder and byproducts from other industries (biofuel, distillery, and palm oil, for instance) onto farms for animal feed [36,37,38]. In climatic zones with severe winters, this coincided with a move from pasture-grazed systems to indoor feeding using cut and carry pasture, either fresh or conserved, and use of concentrates made up of cereal grains, maize, and protein sources reliant on soybean and canola meal [39]. However, despite that trend, pasture-based agriculture is still important in many temperate climatic areas of the world.

Pasture-based agriculture in New Zealand primarily involves introduced grasses sown onto land cleared of predominantly native forest from the late-19th century to mid-20th century [40], and remains the foundation of New Zealand’s economy [16,41,42]. This is unusual for an OECD member country. Ireland is perhaps the closest in comparison with a similar reliance on global markets for generating net wealth, with more than 90% of production from the largest farming sectors of both countries exported [43]. However, for Ireland, agri-food exports are only 9.5% of total exports [44], whereas for New Zealand, agri-food (including forestry and fisheries) makes up 82% of all exports [45]. Pasture-based feeding of ruminants also occurs in some parts of Australia [46], a decreasing area of Europe [47,48,49,50], parts of North America [51], and South America (Uruguay, Chile, Argentina, and Brazil) where native and introduced grassland are used [41,52,53,54,55]. Grazing of largely natural grasslands also occurs in Africa [56] and Asia [57].

Compared with total mixed ration (TMR) feeding, pasture-based feeding of ruminants has been shown to be cost competitive, and result in lower contributions to enteric methane emissions (397 versus 251 g/cow and 200 versus 174 g/kg of milk solids) [58]. In a further study, O’Neill et al. [59] showed that mixing low grass allowance with partial mixed ration resulted in higher methane emission (g/cow/day) than when cows where fed either a high or low grass allowance, by 5.7% and 16%, respectively. However, it was noted that there was no difference between treatments when methane emission was calculated per unit of dry matter intake or milk yield. A completely separate study undertaken in Brazil demonstrated that methane emission was higher (656 vs. 547 g/d) for confined cows solely fed total mixed rations compared with those that received partial total mixed rations and grazed pasture [60]. They concluded that the inclusion of annual ryegrass pasture in the diet of dairy cows maintained animal performance while also reducing enteric methane emissions. Indeed, examining the advantages and disadvantages of grazing over use of TMR feeding shows advantages in animal health and welfare [61,62], improved biodiversity, lower costs, and a positive image of livestock farming [50]. However, TMR does result in the more accurate regulation of dry matter intake and higher milk yield per animal [63], and can effectively allow the incorporation of compounds or extracts that can lead to reduced methane emissions [64,65]. The concern about using these additives relates to health and safety, driven by consumer concerns [66].

In New Zealand, the reliance on pasture-based feed has resulted in seeking improved efficiency gains to remain internationally competitive [67,68]. This quest was driven by the removal of subsidies and price and income support in 1984, when the government was faced with a severe fiscal crisis and implemented an ambitious deregulation programme [69]. Contrary to this, in many other countries animal products, especially those from intensive production systems, are often underpriced [70,71,72] due to government subsidies [73] and the omission of environmental and health costs associated with meat consumption [18].

Milk and meat products from pasture-fed animals have shown superior quality traits to those from animals fed with TMR [74,75]. For example, cheddar cheese made from milk from pasture-fed animals had a healthier fatty acid profile and a yellower colour (because of increased β-carotene content) than cheese made from milk from a TMR-fed animals [76]. Pasture-derived milk has produced butter with improved nutritional characteristics, including significantly higher concentrations of conjugated linoleic acid (cis9, trans-11) and trans-β-carotene than butter from cows fed TMR [77]. Milk from pasture-fed cows (grass and clover) has been shown to have significantly higher concentrations of fat, protein, true protein, and casein, and differing profiles of saturated and unsaturated fatty acid to milk from TMR-fed animals [78,79,80]. This change in nutritional profile is the basis for claims that organic milk is better for health, suggesting that it is pasture that is the key, not organics per se [55,81].

Evidence for the superior nutritional quality of meat, milk, eggs, etc., compared with plant-based foods is seemingly well embedded in the literature [82,83,84,85,86,87,88]. Animal proteins more easily supply essential amino acids and have higher digestibility than plant proteins [85,86]. However, there is a ‘clarion cry’ of concern about the impact of farmed ruminants on water and air quality [7,89,90,91] and welfare [92]. The challenge/opportunity is to show that global issues for animal-sourced foods identified by Food and Agriculture Organization of the United Nations (UN-FAO) [93] are either incorrect (misinformed) and/or manageable through use of targeted technologies or management options.

The UN-FAO issues associated with animal-sourced foods fall in five broad categories:Environment (e.g., deforestation, land-use changes, greenhouse-gas emissions, unsustainable water and land use, pollution, food–feed competition);Herd management (e.g., low productivity, overgrazing, poor animal welfare);Animal health (e.g., diseases, antimicrobial resistance);Human–livestock relationships (e.g., zoonotic- and food-borne diseases);Societal goals (e.g., equity).

Impacts associated with environment are discussed by Caradus et al. [94]. The focus here will primarily be on examining impacts of animal-sourced foods on herd management and animal health. These, along with the environmental impact, have a high technical/scientific component. Human–livestock relationships and societal goals do not, and so there will be minor comment on these aspects. These concerns will be described and then, using pastoral agricultural grazing systems aligned to those found in countries such as New Zealand and Ireland, we will highlight how these farming systems neutralise or mitigate some of the issues. R&D work currently underway to deal with the others will be described. Additional benefits in the efficiency of conversion of solar energy into bioavailable essential amino acids, and specific human nutritional benefits of milk and meat from pasture-raised animals (thus distinguishing them from intensive, confinement systems), will be included in the overall assessment. What must never be forgotten in the goal of feeding people sustainably, is that the biggest threat to the environment is agricultural expansion and concomitant deforestation [95]. To feed an ever-growing global population, producing more food from existing agricultural land is key; reducing yields in developed countries due to ‘land sharing’, without decreasing the need for food, increases imports and so increases harm to biodiversity and natural habitats farther away [96]. When it comes to the global issues outlined by the FAO, all year-round pastoral agriculture grazing systems already perform well but are also taking their responsibilities seriously in addressing environmental impacts by developing science-based technology and systems solutions that are adoptable, adaptable, and scalable.

## 2. Achieving an Optimal Human Diet for Health and Wellness

A recent survey in Australia indicated that about half the population is seeking a healthier diet and that 40% of people frequently consider the question of how healthy or unhealthy a particular food might be [97]. This, however, begs the question of what constitutes a healthy diet, and does that align with public perceptions of what constitutes a healthy diet? It has been observed that “when meat consumption is part of healthy dietary patterns, harmful associations tend to disappear, suggesting that risk is more likely to be contingent on the dietary context rather than meat itself” [98]. Similarly, it has been concluded [22] that full exclusion of meat and dairy products from human diet does “not offer net societal benefit when the environmental benefits were offset against the stringency of actions needed”.

Environmental and animal welfare concerns have been key drivers for pursuing plant-based diets. The emphasis on consuming food for improved health has also contributed to the high profile gained [22,97,99,100]. A comparison of a plant-based diet and a more omnivorous diet would suggest that plant-based diets can meet 2050 global target compositional values for indispensable amino acids. However, when the digestibility of these amino acids in the available foods is considered, there is a diminished capacity for protein synthesis [101,102,103]. Plant-based proteins are densely packed in a rigid molecular structure resulting in low protein digestibility [85,104]. Wolfe et al. [105] proposed a measure to compare different proteins that reflects the percentage of the total daily requirement of the most-limiting essential amino acids that each protein source would deliver if that source was the sole supplier of the total daily protein intake required for a healthy diet. Using this criterion, potato and most animal-derived proteins tend to provide excellent protein quality, whereas soybean and whey proteins are classed as high-quality protein [106]. However, no protein quality claims can be made for gelatine, corn, wheat, hemp, fava bean, oat, pea, canola, rapeseed, lupin, and rice. A flow-on effect of this is that to achieve the essential amino acid intake from some plants may increase consumption of calories and salt [101] and require more cropping land and water [107] with necessary associated use of agrichemicals and fossil fuel [108].

Insects have been identified as another protein substitute for human consumption. Entomophagy, the consumption of insects, has a long history and has been common practice in many cultures [109]. Nutritionally, insects are a source of protein, vitamins, and minerals [110], although the nutritional value can vary between insect species and their stage of development [111]. Even today, entomophagy in some cultures is considered normal and acceptable, while in others it is viewed as unconventional and even repugnant. In most so-called western cultures, insect-based foods are considered unusual and unappealing, resulting in low acceptance [112]. Currently, insect food products are scarce and, in some cultures, promotion of nutritional and health benefits along with assurance of hygiene will be needed to achieve acceptance [113,114]. Sustainability arguments alone will not be sufficient [115,116]. Milk is a high-quality food source [117,118]: “core dairy foods (milk, cheese, yoghurt) have an important role for achieving adequate nutrient intakes in a healthy and lower greenhouse gas emission dietary pattern” [117]. Indeed, “animal-sourced foods are rich in bioavailable nutrients commonly lacking globally, and can make important contributions to food and nutrition security” [86], particularly for the 2.4 billion people who are either moderately or severely food insecure [87]. Milk is the “main contributing food item for calcium (49% of global nutrient availability), Vitamin B2 (24%), lysine (18%), and dietary fat (15%), and contributes more than 10% of global nutrient availability for a further five indispensable amino acids, protein, vitamins A, B5, and B12, phosphorus, and potassium” [6]. Milk provides sufficient vitamin B12, riboflavin and calcium/phosphorus to meet the needs of over 60%, 50% and 35% of the global population, respectively [118].

However, milk is responsible for only 7% of food energy availability. This is balanced by the use of crops on arable land which can meet human dietary energy requirements effectively, but requirements for high-quality protein are met more efficiently by animal production from such land [119]. Coles et al. [119] concluded that “mixed dairy/cropping systems provide the greatest quantity of high-quality protein per unit price to the consumer, have the highest food energy production and can support the dietary requirements of the highest number of people, when assessed as all-year-round production systems”. Further, the FAO indicate that “meat, eggs and milk offer crucial sources of much-needed nutrients which cannot easily be obtained from plant-based foods” particularly during key life stages such as pregnancy and lactation, childhood, adolescence, and old age [93]. Indeed, meat is not just a valued protein source but also an important source of micronutrients and some macronutrients such as phosphorus [120]. It has been considered that “plant and animal foods interact in symbiotic ways to improve human health;” van Vliet et al. [100] contended “that an omnivorous diet rich in whole foods, produced using sustainable agricultural practices that integrates plants and animals in agroecological ways (i.e., in harmony with natural systems), is most likely to benefit human and ecological health”. It is important to note that although van Vliet et al. [100] mentioned “multi-species grazing where there is little dietary overlap such as mixing cattle with goats or sheep…which improves productivity of both animals and vegetation when compared to grazing of either animal alone”, no absolute figures of production or increases were given. The idea of ‘working with nature’ is attractive, but whether it can result in more feed for an increased number of people depends upon the starting point. In New Zealand, attempts to farm ‘regeneratively’, i.e., ‘with nature’ [121], have resulted in decreased production, decreased farm business income, and increased greenhouse gas emission per kg of product [122].

By 2031, it is projected that global food consumption will increase by 1.4% p.a., driven primarily by population growth and mostly within low- and middle-income countries [123]. In high-income countries, a decline in the per capita consumption of sugar and a slowed growth in the consumption of animal protein is predicted, resulting from concerns about health and the environment. However, in middle-income countries, greater total food consumption and an increased diversity of diet, with growing shares of animal products and fats, is projected to occur. Overall, this could lead to a 1.5% annual growth in livestock and fish production, but most is projected to result from improvements in per-animal productivity due to more efficient herd management and higher feed intensity. Having said that, more than half of the global growth in meat production is predicted to be in poultry due to sustained profitability and favourable meat-to-feed price ratios. Agricultural greenhouse gas emissions are projected to grow at a lower rate than production, due to yield improvements and a reduction in the share of ruminant production. Emissions from agriculture are projected to increase by 6% through to 2031, with livestock accounting for 90% of this increase [123]. To mitigate against this increase, average global agricultural productivity would need to increase by 28% during that period. This would mean a 24% increase in average global yields of crops, which is close to double the increase achieved over the past decade (13%), and a 31% increase in global animal productivity which is much greater than the growth recorded during the last decade. As noted in the Introduction, global rates of gain in agricultural productivity appear to be static or declining, with a suggestion that in recent decades “global agriculture has grown more vulnerable to on-going climate change” [33]. Increased agricultural investment and innovation will be essential to ensure sustainable productivity growth.

Further, to feed the projected global population growth by 2050, it is estimated that meat production will need to double from the current level of production [124]. This will be a challenge using current production systems, but does provide the opportunity of making meat differently via alternative proteins, which might come from insects, fungi, bacteria, or algae, or through cultured meat [125]. However, the latter alone is unlikely to satisfy the need, since it has been estimated that to supply just 10% of the world’s meat consumption (i.e., 40 million tonnes) from laboratory cultivated meat would require 4000 ‘factories’, each with 130 bioreactor lines, each having 10,000 L bioreactor tanks [126] implying high capital and operating costs and heavy energy consumption. There are still significant technical, ethical, regulatory, and commercial challenges to getting these “cell-based meat” products widely available in the market and may only be available to the more affluent sections of society [127], and actually be able to replace conventional meat rather than become a luxury item [128]. Cultured meat production costs have been shown to be 10,000 to 100 higher than benchmark values for comparable traditional meat products [129]. The significant scaling-up problem associated with the technology means that demand for premium grass-fed protein is likely to remain or even grow as demand for protein increases [130]. Additionally, it is considered by nearly 60% of the population surveyed in Australia that so-called healthy eating would cost more [97], suggesting that more cost-efficient production systems will be needed.

## 3. Animal Perspectives: Managing Herds, Animal Health, and Zoonoses

The challenge/opportunity for animal-sourced foods is to demonstrate that global issues identified by the UN-FAO are either misinformed and/or are manageable through use of specific technologies or management. The most significant issue identified by the FAO is the impact of food production from livestock on the environment is discussed in Caradus et al. [94]. However, there are important issues also arising from the interactions between animals and their environment (land, soil, and plants), challenges of managing animal health in the relatively uncontrolled setting of free-range grazing of (generally) multi-species swards, and the risk of disease transfer between animals and their human managers/carers which are discussed below.

### 3.1. Herd Management

The key issues identified by the FAO [87] in relation to herd management included overgrazing, low productivity, and poor animal welfare. The first two of these will be addressed together since they are closely inter-related and the solutions to them are based on common biophysical and ecophysiological principles and, in some instances, closely related management practices. Animal welfare outcomes are generally related to factors such as levels of feeding, or provision of adequate shelter and protection from heat stress and other direct environmental variables which are well covered elsewhere.

Adaptive herd management is critical to achieving several of the 17 UN Sustainable Development Goals (e.g., 2—Zero Hunger, 6—Clean Water, and 15—Life on Land) for several reasons, among which controlling soil erosion is of prime importance. Soil erosion is one of the most significant threats to global food productivity since eroded soils commonly have lower nutrient status, low soil C and lower water holding capacity compared with uneroded soils, and they support poorer crop and pasture/grassland biomass production [131,132]. Historically, land use change in pursuit of economic development has been the major driver of accelerated rates of erosion. This land use trend continues today, as noted by Caradus et al. [97]. As a result, Borrelli et al. [133] estimated that total global soil erosion by water increased between 2001 and 2012 (from 35.0 to 35.9 petagrams (Pg) per year). Half of this increase was from cropland (at an average intensity of 12.7 metagrams (Mg)/ha/year; cropland occupied only 11% of the total land mass) whereas most of the rest came from seminatural vegetation (average of 1.84 Mg/ha/year). Estimated forest erosion intensity was 0.16 Mg/ha/yr. The differences in erosion intensity are inversely proportional to the relative mass, duration (over an annual cycle), and type of vegetation cover among the three land use classes: forest >> seminatural vegetation >> cropland. The global increase in soil erosion of 0.9 Pg/year between 2001 and 2012 was largely a result of the net increase in cropping land of 0.22 m km^2^ over the same period [133].

The deleterious effects of overgrazing, soil tillage, and other agricultural practices with negative effects on soil stability and structure are well documented [134,135]. However, it is difficult to disentangle these completely from factors such as rainfall variability or drought, and social forces embedded in different cultures around the world. Overgrazing in some parts of the world, such as sub-Saharan Africa, can cause extensive degradation of the vegetation community, associated loss of biodiversity [136] and increased soil erosion [137]. Food security is a pressing issue in such regions, and the conversion of seminatural vegetation to crop cultivation to supply more human-edible food means that stocking rate increases on the reduced area of grazing land, placing greater defoliation pressure on the cornerstone plant species in the grassland [138]. When drought strikes, there is often little capacity to de-stock to protect valuable plant species, rapidly under-mining the resilience of the grassland community.

Where there is greater knowledge of the ecology of seminatural vegetation communities and more resources to support management decisions (including the herd manger skill level and access to information), it is possible to avert over-grazing during drought, sustain grassland productivity and control erosion to a substantial degree. The mixed woodland–grassland vegetation of arid western, central, and northern Australia is an example of fragile vegetation communities subject to frequent intense and prolonged droughts. These communities carry 60% of Australia’s national beef herd of 24.4 M animals, contributing about 2% of global beef meat production [139]. The rangelands have suffered a series of severe droughts over the past hundred years, leading to episodic overstocking and subsequent depletion of browse vegetation leading to increased soil erosion, dominance of woody species, and decline in forage availability [140]. In response, the concept of a ‘safe’, or long term, livestock carrying capacity (LTCC) has been developed [141,142] to ensure the pasture resource could be sustained for 30 years or more under the pressure of decadal variation in rainfall. The LTCC is defined as “the number of animals (e.g., dry sheep equivalents or beef equivalents) that can be carried on a land system, paddock or property without any decrease in pasture condition and without accelerated soil erosion” [143]. The LTCC is a strategic tool to help graziers manage individual land units based on the vegetation type, local climate, and other resources. The experience of graziers themselves in dealing with drought was incorporated in the development of the concept. In Queensland, graziers can now access property-specific information from the online system FORAGE [144] which draws data from spatial datasets of climate, soils, woody vegetation cover, and satellite-derived ground cover [141,142,145]. The FORAGE tool is a culmination of the “comprehensive synthesis of the advances in grazing land sciences and modelling over the last couple of decades”, [142] and a good example of the potential to implement practical solutions based on sound science made accessible to all end users via the internet.

The FORAGE tool also provides a basis for forecasting the impacts of future climate change on the rangeland beef industry. In a systematic simulation of the effects of climate change on forage production, McKeon et al. [146] concluded that “decreases in LTCC [are anticipated] given that the ‘best estimate’ of climate change across the rangelands is for a decline (or little change) in rainfall and an increase in temperature”. The capacity to simulate likely future outcomes of climate change is critical to developing policy and management responses that will ensure sustainable use of the 2.5 billion ha of land used by livestock [29] globally. Across the three major global land use classes of forest, seminatural vegetation, and cropland, Borrelli et al. [147] identified “a trend toward a more vigorous hydrological cycle, which could increase global water erosion (+30% to +66%) [above current total global erosion rates of 36 Pg/year]”. This would represent a large reduction in food production capacity with obvious flow on impacts for food security. This would be especially the case in Asia and the sub-continent, sub-Saharan Africa, and Latin and Central America where climate-induced acceleration of soil erosion is predicted to increase at the highest rates and potentially exceed the capacity of technologies such as conservation agriculture to offset the worst effects [147].

Moving to the intensive, high rainfall temperate pasture systems of New Zealand and Ireland, over-grazing has a different context and meaning, and the consequences of over-grazing are far more benign than those experienced in regions covered by seminatural vegetation. Here, pastures generally provide complete vegetative cover of soils, support root mass at least equal to above-ground biomass and sustain soil organic carbon contents between 3% and 5%. ‘Over-grazing’ in this context typically means a reduction in leaf area index below that required to maximise C assimilation, and so results in loss of potential herbage yield. Based on an extensive scientific knowledge base of plant growth processes and the mass flux of C in temperate C_3_ grass species, comprehensively outlined by Parsons [148], and now extending into C_4_ grass species [149], grazing decision rules [150] have been developed to optimise the balance between maintaining herbage mass for C assimilation versus achieving high rates of herbage intake for animal production. In temperate perennial pastures, maintaining total herbage mass between 1500 and 3000 kg DM/ha using rotational grazing strikes an optimal balance between the needs of the plant and the needs of the animal (the latter including digestible feed of approximately 11 MJ/kg DM metabolisable energy content and at least 18% crude protein). Rotational grazing allows managers to allocate feed according to herd requirements, on a daily basis at the more-intensive end of the scale of management sophistication. The success of the New Zealand and Irish livestock industries in producing large amounts of high-quality animal protein food per hectare is founded on a large body of pasture, animal, and systems science [151] facilitated by relatively mild temperatures and consistent rainfall, augmented by some irrigation in New Zealand. These industries export much of the food they produce to other countries: for example, it is estimated that New Zealand produces enough animal protein to feed 45 million people per year [152]—but has a national population of only about 5 million people.

Viewed in the global context of food production challenges, sustainability problems confronting these industries are relatively minor. Greenhouse gas and nutrient emissions must be addressed, as discussed in Caradus et al. [94]. Climate change will impact pasture growth rates, mostly negatively [153,154], but may be potentially positive in some situations [155]. Associated with negative impacts of warmer, drier climate on pasture growth rates, the longevity of the traditional perennial ryegrass-white clover pasture type is also expected to decline resulting in higher rates of pasture replacement [156]. Indeed, the latter impact is already evident across the upper North Island of New Zealand [157] bringing higher costs of production but, importantly, also increased risk of nitrate leaching [158] and high rates of loss of soil C [159]. Adaptation will be required but the R&D response is lagging as pasture research now sits well down the list of Government and industry stakeholder priorities [160].

### 3.2. Animal Health

Some grazed forages can cause animal health and welfare issues. These can include fescue toxicosis, ryegrass staggers, bloat, facial eczema, grass tetany, nitrate, and other toxicities. However, for each of these there are management solutions that can reduce or eliminate the issue (Table 1). Other adverse effects from some forage plants can be caused by phyto-oestrogens and cyanogenic glycosides in clovers, s-methylcysteine sulphoxide associated with brassicas, and alkaloids such as pyrrolizidine in Phalaris [161].

However, practical cost-effective solutions are available to mitigate these animal health and welfare issues, as described in Table 1.

Antimicrobial drug use in farm animals has received attention globally because of the potential for development of antimicrobial resistance, including in humans. Evidence indicates, however, that misuse in humans is the main driver for emergence and persistence of antimicrobial resistance in humans [192]. New Zealand has a low use of antimicrobials in food producing animals compared with other developed countries, but ‘there is room for improvement’ [193]. The low use reflects the fact that animals are not in feedlots at high stocking rates, and not fed growth promoting substances. Illustrating the difference for the milking herd, in the UK, where animals are commonly housed through late autumn, winter, and spring, 43% of the animals will experience mastitis and require antibiotic intervention annually whereas in New Zealand, where animals are at pasture all year round, the figure is 20% [194]. In the spirit of ‘room for improvement’, New Zealand Food Safety [195] reported a 23% decrease in sales of agricultural use antibiotics between 2021 and 2022. Total antibiotic sales in intramammary products (to control mastitis) in 2022 were 11% less than the peak recorded in 2017.

### 3.3. Human–Livestock Relationships

Zoonotic pathogens that cause food-borne disease worldwide include *Campylobacter*, *Salmonella*, *Listeria*, *Staphylococcus*, *Brucella*, *Clostridium*, *Mycobacterium*, *Colibacilus*, and *Escherichia coli* [196,197]. Because no effective interventions have eliminated them from animals and food, most of these pathogenic bacteria will continue to cause outbreaks and deaths throughout the world [196]. Contamination of animal products with microbial pathogens can occur in the field, at slaughter and during processing [198]. Control of Shiga toxin-producing *E. coli* has been sought, but with varying results, through vaccination, treatment with probiotics, administration of bacteriophages, and modification of the diet [199]. A meta-analysis has shown that feedlot cattle have a consistently higher prevalence for carrying *Campylobacter* spp. compared with adult cattle on pasture [200]. High stocking rates and close interaction between animals and humans facilitates disease transfer.

Increased interaction with wild animals following expansion of grazing near forests has been flagged as a driver of zoonotic spillover—the transmission of pathogens from wild animals to humans. The accelerated growth of the world population and the increased loss of global biodiversity, as a result of human activity, has led to the suggestion that spillover events will become more frequent [201]. Of importance is that it is also possible for wild animals to be infected by farmed animals [202].

## 4. Additional Benefits of Animal-Sourced Foods from Pasture-Based Systems

The additional benefits from animal foods have been reviewed in detail for edible (at least in theory) and inedible products [203,204,205]. Far beyond wool and leather, the possibilities include horn, bone, and antler, plus improving understanding of the use of offal. In Germany, it has been estimated that eating organ meats twice a week could reduce greenhouse gases from livestock by 12.5%, by reducing waste and hence both landfill and the need for more animals to feed people [206]. This reduction has been calculated to be equivalent to taking 214 million cars off the road for good [206]. Additionally, the use of offal and some organ meats in preparation of pet food can also lead to minimization of waste [207,208].

For wool, protecting the animal with a coat improves the quality of the fleece (e.g., Australian Wool Education Trust [209]; it is likely that free-range animals on pasture do not achieve the same quality as housed animals. Leather is also likely to be improved with sheltered living.

A recently developing opportunity associated with farming deer is the health benefits identified from consumption of deer velvet. Human clinical trials for a prostate function extract “showed good results especially for improved urinary flow rate, for dosages of 500 milligrams and 1000 mg per day, when compared to a placebo group over a period of 12 weeks. The dosage was also well tolerated”. Trials with mice on treadmills with or without extracts from velvet, indicated alleviation of symptoms of fatigue in mice as an effective anti-inflammatory agent [210].

In July, the Korean Ministry of Food and Drug Safety approved the prostate function and anti-fatigue claims made by Kwangdong (a health food company), which were backed by peer-reviewed results. As a result, Kwangdong can now use these claims in their new products launching next year [211]. New Zealand was a pioneer in farming deer, allowing high quality products to be harvested with all regard for food safety for venison and velvet, and high animal welfare because feeding on pasture can be managed to match animal requirements.

## 5. Concluding Comments and Looking to the Future

The real or perceived connection between livestock agriculture and environmental issues has led to two main propositions—(1) ‘sustainable intensification’ which encourages ecological responsibility to reduce the ‘ecological hoofprint’ [212]; and (2) growing calls to ‘de-animalise’ the food system [213], and in doing so, dramatically reduce the global population of cattle through shifting to manufacture of ‘plant-based’ or ‘lab-grown’ meat and dairy alternatives. Calls to radically reduce or eliminate animal protein production in response to the second of these propositions are misplaced because they allow no room for the development of solutions to the core challenges of reducing ruminant methane emissions, nutrient contamination of freshwater, and soil erosion—of which there are many, as outlined in this paper and by Caradus et al. [94]. In reality, the two propositions need not be mutually exclusive if globally integrated investment in the critical science disciplines is forthcoming, and the technological and knowledge outcomes are evenly shared among the world’s grassland farming industries/communities.

Similar calls to remove animal protein foods on the grounds of adverse human health outcomes are mistaken, as the ample evidence for the benefits of such foods reviewed in this paper demonstrate. Removing such a significant source of nutrition from food supply chains will surely intensify the challenge of providing adequate nutrition for the growing global population. This is especially so when we consider recent evidence that anthropogenic climate change has reduced rates of global agricultural total factor productivity growth by ~20% over the past six decades and increased the vulnerability of current food production systems to future climate change [33].

The real challenge for the future, therefore, is raising productivity on the land currently used for both crop production and livestock production. The alternative is further expansion into non-agricultural areas to provide for the increased demand in human-edible feed materials from both plants and livestock [29]. Solutions to this projected dilemma is to reduce food wastage, and improve global trade opportunities, but equally importantly to work to improve productivity (yield per unit area of land). In addition, this must all be carried out with a reduced impact on the environment and detrimental effects on land, water, and air. Livestock system must be managed within the carrying capacity of the land to ensure that water (both quantity used and quality resulting) is acceptable.

The advantages of using grazed pasture for ruminant meat and milk production:Animal-sourced foods contain essential amino acid and micronutrients and almost always have a higher digestibility than plant proteins;Greater use of pasture to feed animals could possibly result in less use of food that could be used for human consumption, e.g., maize, soybean, and cereal crops;Compared with total mixed ration feeding, pasture-based feeding of ruminants has been shown to be not only cost competitive but can also result in lower contributions to enteric methane emissions;Ruminants can provide food from land otherwise unable to be used for cropping;Ruminants are able to consume feed that cannot be eaten directly by humans;Ruminants provide more than just food—they also provide leather, fibre etc.

While some forages grazed by ruminants can cause animal health and welfare issues, many of these are now well understood and mitigation strategies are available.

Animal-sourced foods are an important part of the human diet, and while some unintended consequences associated with animal health and herd management have occurred, technologies and systems to provide solutions to these are available and under refinement.

## Figures and Tables

**Table 1 animals-14-01077-t001:** Solutions that will mitigate the likely impacts of some forages, and the grazing of forages on animal health and welfare.

Animal Health and Welfare Issue	Cause	Solution	Reference
Fescue toxicosis and heat stress	Ergovaline expression by *Epichloë* in grass	Use of *Epichloë* strains that do not produce ergovaline; incorporate legumes	[162,163,164,165,166,167]
Ryegrass staggers	Lolitrem B expression by *Epichloë* in grass	Use of *Epichloë* strains that do not produce lolitrem B	[167,168,169,170,171]
Facial eczema (also causes liver damage)	*Pseudopithomyces chartarum* producing sporidesmin	Avoid grazing pastures with excessive dead leaf material in late summer autumn; spray pastures with fungicides and treat animals with zinc	[172,173]
Toxins associated saprophytic fungus *Fusarium*	Proliferate in cooler temperatures on perennial ryegrass	Ensure high clover (close grazing in late spring) content and switch to tall fescue	[174]
Bloat	Consuming vegetative legume pastures such as clovers and alfalfa which cause gas in rumen faster than it can expelled	Ensure the legume component is less than 50%; include non-bloating legume like Cicer milkvetch, sainfoin or birdsfoot trefoil; feed an anti-foaming agent; feed grass hay	[166,175,176]
Nutritional haemoglobinuria and glucosinolates (thought to be goitrogenic), and increase photosensitivity	Forage brassicas	Provision of adequate shade, injection of analgesic and anti-inflammatory drugs, and application of zinc-containing ointments and balms	[177]
Laminitis or lameness	Associated with highly fermentable diets	Design of laneways, feed pads and dairies	[178]
Grass tetany	Low blood levels of Magnesium—often associated with early spring grazing of succulent cool-season grasses	Delay spring grazing; feeding a legume hay with spring grass pastures since legumes are. higher in magnesium than grasses; and providing a mineral supplement	[166]
Prussic acid toxicity which causes salivation, laboured breathing and muscle spasms	Toxin that occurs in annual grasses such as Johnsongrass, sorghum and sorghum-sudan hybrids when stressed due to drought or frost	Avoid grazing a week after the end of a drought or killing frost; use pearl millet as a warm-season annual forage (does not produce prussic acid)	[166]
Nitrate poisoning	Excess nitrates will accumulate in the lower stems of when some plants are stressed by drought, heavy rain or long period of cloudy weather	Do not graze suspected forages several days after the end of a drought; chopping forage and diluting with clean hay; minimizing nitrogen use	[166]
Hypomagnesaemia and milk fever	High concentrations of potassium in pastures	Supplement with magnesium oxide and rumen modifiers	[179]
Ketosis	Response to low glucose availability resulting in elevated levels of ketones in blood or urine	Oral drenching of propylene glycol; use antibiotics lasalocid and monensin	[180,181,182]
Acidosis—large quantities of Volatile Fatty Acids and lactic acid act to lower rumen pH to non-physiological levels	Occurs in pasture-based dairy systems where cows are fed supplements; also on pastures low in Neutral Detergent Fibre and higher in sugars	Use Neutral Detergent Fibre in pasture-based diets that are higher than 30% with 75% of supplied by coarse forage; use antibiotics tylosin and virginiamycin	[178,183,184]
Endoparasites—such as nematodes	Parasite resistance to anthelmintics is well documented	Use of medicinal plants; breeding of worm resistant sheep; specific grazing strategies for worm control	[185,186,187,188]
Ectoparasites such as flies (causing fly strike) and mites	Resistance to synthetic neurotoxic insecticides, environmental contamination, and effects on human health	Biological control, off-host trapping systems and selective treatment of susceptible individuals	[189]
Mastitis	*Escherichia coli* infection of udder tissue; can have antimicrobial resistance to all major antibiotic classes	Good herd health management and environmental hygiene; vaccination; use steroidal or non-steroidal anti-inflammatory drugs; use of antibiotics for severe infections	[190]
Fractured humerus condition	Affects first- and second-lactation spring calving dairy cows	No known cost-effective management solutions	[191]

## Data Availability

Not applicable.

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
