# Peer review of "Improving Human Diets and Welfare through Using Herbivore-Based Foods: 1. Human and Animal Perspectives"

_animals, 2024, doi:10.3390/ani14071077_

Round 1

Reviewer 1 Report

Comments and Suggestions for Authors

 The article concerns a very important issue: the relationship between livestock farming and environmental issues. The authors approached the topic from many sides, which greatly enhances the value of this review article. The selected methods based on the analysis of the modern literature on the subject seem to be sufficient. This is a very important voice in the ongoing discussion on the future of animal production. It is often influenced more by ideology than by facts and scientific research. The article provides an overview of research findings in which practitioners have often been involved. Therefore, its results should be read not only by politicians deciding on such an important economic sector as agriculture and the food industry, not only in New Zealand, but also by society. Especially young people who adopt a vegetarian diet do not have knowledge about the beneficial effects of food of animal origin, for example milk, on human health. Perhaps you could consider expanding the thread on the potential effects of protein substitutes such as eating insects. It is indicated that time and research are needed to determine the effects of eating them on human health.

Best regards

Reviewer 2 Report

Comments and Suggestions for Authors

The paper deals on a topic of peculiar interest, the affirmation of the importance of animal-source foods in the human diet animal compared with plant-only based food examining in a review different papers in terms of quantification and mitigation of unintended consequences associated with environment, animal health and herd management.

Please check comments and indications in the attached file.

The paper is well written and needs only of some minor interventions 

Author Response

The paper deals on a topic of peculiar interest, the affirmation of the importance of animal-source foods in the human diet animal compared with plant-only based food examining in a review different papers in terms of quantification and mitigation of unintended consequences associated with environment, animal health and herd management.

Please check comments and indications in the attached file.

The paper is well written and needs only of some minor interventions.

Author response

In response to the request to check comments and indications in the attached file, these have been checked and suggested changes and additions acted upon. See attached file with comments.

Reviewer 3 Report

Comments and Suggestions for Authors

This is a well-written and heroic attempt to review an enormous and complex subject. I have only two suggestions.

1. p3 O'Neill et al. It is too simplistic to conclude from this paper that pasture-based diets will reduce methane emissions relative to that of cows fed TMRs in confinement. The stoichiometry of TMR digestion is to increase propionate:acetate ratio relative to pasture. Moreover, milk yields on TMR are much greater thereby increasing the proportion of ME directed to yield relativr to that for maintenance. I advise more caution in discussing the greenhouse gas impact of cows. It needs at least a book.

2' Perhaps the major factor in support of the sustainability of diets based on pasture and conserved non-cereal products is the extent to which food for human consumption can be derived from complementary rather than competitive plant sources, There is good evidence from the UK that lactating cows fed such diets can yield at least 30% more energy and protein for human consumption than they consume from sources that could be fed directly to humans.

Reviewer 4 Report

Comments and Suggestions for Authors

General comments: This manuscript provides a review that documents some of the advantages and limitations of animal production systems as important food sources for humans with emphasis on pasture-based production systems.  They conclude that animal products are very important food sources and that other animal products are important for clothing, etc. They point out unintended consequences such as effects on the environment including greenhouse gases, water quality, and effects of overgrazing. They also note forage-related issues with animal health. The manuscript also includes references that point out ways in which such environmental issues and animal health concerns can be mitigated. The manuscript is generally well written with some mostly minor areas of concern which are noted in specific comments. One area of writing style that could be improved is the use of numerous phrases set off by parentheses. Perhaps the flow would be improved by rewording those phrases into extra sentences to more clearly express the point(s) to be made. In the section on animal health, perhaps a mention of a growing concern about fractured humerus bones in young NZ dairy cows which may be indicative of mineral imbalances in pasture forages for young lactating cows that are still growing. See Animals 2024, 14(3), 524; https://doi.org/10.3390/ani14030524.

Specific comments:

L 2: In title use a period after 1: “1.”

L 27: Use “whereas” rather than “while” Also lines 71, 244,366

L 48: Use comma after: “… 2022), …”

L 57: “Worldwide, …”

L 72 and elsewhere: Periods and commas should be within quotation marks within or at end of sentences: “ … in low emissions.” Also lines 86, 88, 93, 229, 271, 280, 417, 423, 492, 534, 1048.

L 81-83: Reword for clarity: Although this is shown as a quote, what is meant by “… would produce substantial negative emissions of all three …” I assume “negative” means higher emissions?

L 97: Awkward use of the inserted phrase. Reword lines 95-100 accordingly for improved flow.

L 99: Perhaps “quantity” rather than “quantum?”

L 377: “ … Overgrazing in some parts …”

L 483: Reword, perhaps: “… however, misuse of antimicrobial drugs in humans and not animals is the main …”

L 519-524: Perhaps consider the use of offal and some organ meats in preparation of pet foods in this section to minimize waste.

Comments on the Quality of English Language

My understanding of punctuation is that periods at the end of quoted sentences or commas within a quoted phrase should be inside the quotation marks. I note that British spellings are used which is fine if the journal prefers or accepts those.

Round 2

Reviewer 2 Report

Comments and Suggestions for Authors

I appreciated all the interventions to improve the paper.

I would suggest substituting the citation Renna et al. (2012) [79] related to dairy goats with the one "Renna M, Ferlay A, Lussiana C, Bany D, Graulet B, Wyss U, Ravetto Enri S, Battaglini L M and Coppa M (2020) Relative hierarchy of farming practices affecting the fatty acid composition of permanent grasslands and of the derived bulk milk. Animal Feed Science and Technology 267 114561 (https://doi.org/10.1016/j.anifeedsci.2020.114561)" concerning dairy cows and, more pertinent and updated (2020)

Reviewer 3 Report

Comments and Suggestions for Authors

This is a well researched and well thought out review. You have taken account of the issues raised by the referees  relating to the first draft and it is now, in my opinion, acceptable for publication.

Reviewer 4 Report

Comments and Suggestions for Authors

The manuscript has been edited and improved as requested.
